# Is there a difference in thresholds for revision between shoulder arthroplasty types? A National Joint Registry Study

Olivia O'Malley[1]*, Andrew Davies[2], Amar Rangan[3], Sanjeeve Sabharwal[2], Peter Reilly[1]

1 Department of Bioengineering, Cutrale Perioperative & Ageing Group, Imperial College London, London, United Kingdom, 2 Department of Trauma & Orthopaedics, Imperial College Healthcare NHS Foundation Trust, London, United Kingdom, 3 Hull York Medical School & Department of Health Sciences, University of York, York, United Kingdom

* o.omalley23@imperial.ac.uk

## Abstract

### Introduction

Shoulder arthroplasty procedures have increased significantly, with reverse shoulder arthroplasty (RSA) becoming more common. While RSA revision rates are reported as low, these figures may not accurately reflect implant success. Factors such as older patient demographics and surgeon reluctance to perform complex revisions may contribute to lower revision rates. This perception may encourage broader RSA use in younger patients, potentially increasing long-term revision burdens. This study examines whether revision thresholds differ between shoulder arthroplasty types to determine if low revision rates reported signify true implant success.

### Methods

All shoulder arthroplasties from the 1st April 2012 to the 31st March 2022 were requested from the National Joint Registry (NJR). Mean postoperative Oxford Shoulder Score (OSS) was calculated for RSA, total shoulder arthroplasty (TSA), and hemiarthroplasty (HA). Revision rates were analysed between implants for patients with a mean postoperative OSS of <29 (a pre-defined unsatisfactory score) and those with the lowest 25%, and lowest 10% of OSS scores. Chi-squared tests with Bonferroni correction assessed differences among implant groups.

### Results

Among 21,918 NJR patients with postoperative OSS data, HA had the highest proportion of patients with 'unsatisfactory' function (38.12%), followed by RSA (26.99%) and TSA (15.35%). 4.87% of RSA patients with unsatisfactory function were revised, significantly less than TSA (10.58%) and HA (13.86%) (p < 0.001). In those with the lowest 25% of OSS scores, revision rates were 4.78% for RSA, 8.76% for TSA, and

**Data availability statement:** The data for this study were obtained from the National Joint Registry for England, Wales, Northern Ireland and the Isle of Man but legal restrictions apply to the availability of these data, which were used under license for the current study. Permission to access the data can be sought from the NJR Research Sub-committee (http://www.njrcentre.org.uk/njrcentre/Research/Researchrequests/tabid/305/Default.aspx). For information regarding data requests, please contact the NJR Research & Governance Manager Alyson.Ottley@njr.org.uk.

**Funding:** The authors disclose a receipt of the following financial or material support for the research, authorship, and/or publication of this article: an institutional British Elbow and Shoulder Society pump primer research grant, and a Royal College of Surgeons Research Fellowship. The funders had no role in study design, data collection and analysis, decision to publish, or preparation of the manuscript.

**Competing interests:** No competing interests. O.O'Malley is an RCS England Research Fellow funded by The Arthritis Research Trust Research Fellowship with support from the Rosetrees Trust. A. Rangan reports an institutional grant from NIHR and AO UK&I, as well as research and educational grants from DePuy J&J Ltd, unrelated to this study. A. Rangan is also an Elected Trustee of the British Orthopaedic Association and is on the funding committee of NIHR i4i.

15.02% for HA (p < 0.001). In the lowest 10%, revisions occurred in 6.53% of RSAs, 12.44% of TSAs, and 17.03% of HAs (p < 0.001). No significant difference was found between TSA and HA (p = 0.06).

## Conclusion

RSA has lower revision rates than HA and TSA; however, this may not reflect superior implant performance. Patients with poor RSA function are less likely to undergo revision, suggesting higher revision thresholds. As RSA use expands, its assumed low revision rate must be reassessed to prevent long-term burdens of poorly functioning implants. Further research is needed to determine whether surgical selection bias influences revision rates and to establish additional benchmarks or surrogates in joint registries for a more comprehensive assessment of implant performance.

## Introduction

Shoulder arthroplasty has been shown to provide substantial pain relief and functional benefits for patients with elective indications such as primary shoulder arthritis, rotator cuff arthropathy (CTA) and following traumatic injuries to the shoulder [1]. The use of shoulder arthroplasty is increasing with 8221 cases recorded in the National Joint Registry (NJR) in 2023 compared to 2764 in 2012 when the registry was established [2]. There are three types of shoulder arthroplasty, Total Shoulder Arthroplasty (TSA), Reverse Shoulder Arthoplasty (RSA) and Hemiarthroplasty (HA). In 2012 the most common implant was HA which was undertaken in 37.3% of cases, by 2014 this had dropped to 25.0% with TSA undertaken in 28.5% of cases and RSA in 35.5% [2]. This trend has continued, and in 2023 only 4.7% of cases were HA, 19.8% were TSA and RSA use had dramatically increased in use to 63.4% of implants [2]. The increasing use of RSA may be attributed to its increased scope of use or perceived success due to low reported revision rates. RSA was originally primarily designed for CTA, however it is now being used for a breadth of indications, both electively and for trauma, as well as in broadened patient demographics [3]. The revision rates for the majority of these indications are reported to be low and the 10 year revision rate for elective indications reported in the the NJR for RSA is 4.41%, in TSA is 6.42% and in HA is 10.87% [2,3].

Revision rates are a marker of success however they do not reflect failing implants that do not go on to revision. The RSA has traditionally been implanted into an older patient demographic that may be more accepting of functional decline and surgeons may be reluctant to undertake technically demanding revision procedures in this high-risk cohort. Therefore, the low revision rates reported for RSA may not accurately reflect the true success of the implant or reflect patient function post operatively. This perception of lower revision rates may, in turn, encourage increased use of the implant in younger patients or for indications beyond its original design. This could potentially lead to a greater long-term revision burden as these implants fail over time which may put significant pressure on health care systems. This increase in

revision burden is already being seen in the United States at an annual cost of $205 million [1]. This study aims to test the hypothesis whether there is a different functional threshold to revise between the differing implants and so to determine whether low revision rates are truly a marker of implant success. The study will use Patient Reported Outcome Measures (PROMS) collected by the NJR in the form of post-operative Oxford Shoulder Score (OSS) as a marker of functional threshold.

## Methods

### Data source

Data was requested from the NJR for all shoulder arthroplasty cases between 1st April 2012–31st March 2022. Patients are consented pre-operatively for their inclusion in the registry. The databases included a primary data file with patient demographics, implant data and outcomes in terms of 'Revised' 'Unrevised' or 'Death'. Implants were validated against their coded implant types (TSA, RSA, HA) by cross checking with implant components recorded in the NJR database to ensure accurate coding had been completed. The NJR also collects PROMS in the form of OSS score which was also requested for this time period. OSS is a patient focused, shoulder specific questionnaire consisting of 12 questions which are scored from 0 to 4 with a maximum score of 48 [4,5]. The score contains questions pertaining to patient function as well as pain and a higher score reflects a better outcome. The score has been extensively tested and validated and is deemed a reliable test for assessing patients shoulder symptoms [6]. Within the NJR, OSS is recorded pre-operatively, at 6 months, 3 and 5 years. As defined by Dawson et. al within the questionnaire if there are up to two items missing, the average of the remaining items can be substituted for the missing values and if more than two items are missing, the results will be disregarded [5]. This method of substitution, where up to two answers were missing, was implored in this study and a total OSS score was calculated for each patient at each follow up. If a patient had more than two answers missing they were excluded from the study as a total OSS score could not be calculated. The primary outcome database was linked with the PROMs database using primary NJR index number which is a unique identifier for each patient.

### Outcomes

The primary outcome was to compare at differing OSS score cutoffs, the percentage of patients with each implant type who went on to have revision surgery. The secondary outcome was to compare the demographics and surgical characteristics of those patients that went on to have a revision between implant types to determine if there is a possible channelling bias when selecting patients for revision.

### Statistical analysis

For each patient with a validated total post operative OSS score, a mean OSS score (from the 6 month, 3 year and 5 year follow up) was calculated. For the primary outcome analysis, three cut off OSS scores were used to compare the percentage of patients receiving a revision for each implant. The first cut off was an OSS score of 29 which has been defined in the literature as an OSS score predictive of patient satisfaction [7]. The 1st quartile (lowest 25%) and the 1st decile (lowest 10%) mean postoperative OSS score rounded to the nearest whole number were used as the other cut off scores. These secondary cut off scores were used as an alternative marker of poor function as it is known that post-operatively the different implants have differing functional outcomes [2]. This analysis therefore enables comparison of poorly functioning implants taking into account differing expected post operative function for that specific implant type.

Demographics were compared between implant groups at each OSS cut off. Age at primary arthroplasty was compared using Kruskal Wallis H test due to skew of the data. Pairwise analysis was completed using Dunns Test with Bonferroni adjustment to control for the risk of type 1 error due to multiple testing. Categorical outcomes such as American Society of Anaesthesiologist Physical Status Classification (ASA) & gender, were assessed by Chi Squared test. Pairwise comparisons between groups were completed if the overall Chi squared showed a significant difference (p<0.05). Bonferroni adjustment was used to account for

multiple comparisons with a significance level deemed at p<0.0167. The percentage of patients within each implant that went onto to be revised at each OSS score cutoff were calculated and presented descriptively as a percentage. The percentage of those revised was compared between groups using a Chi Squared test. Pairwise analysis was completed to compare the different implants with Bonferroni adjustment. As the implants were designed for different primary indications, for example TSA is sparingly used in trauma, a sensitivity analysis for revision percentage was completed in just elective circumstances.

For the secondary analysis, demographics were compared between those that went on to revision between implants with the same method as the primary outcome analysis. The three most common revision indications were identified and displayed as percentages of revision cases, to note the indications for revision in the NJR data collection are not mutually exclusive. Shoulder function in the form of OSS score in the most recent follow up prior to revision was assessed using the Kruskal – Wallis H test and a Dunn's test with Bonferroni adjustment was used to identify which specific groups differed. The statistical analysis was performed using StataSE v16 (StataCorp, USA). The STrengthening the Reporting of OBservational studies in Epidemiology (STROBE) guidelines were adhered to in this study [8].

## Results

There were 67920 validated implants recorded within the NJR between March 2012 and April 2022. 21918 patients had a post operative OSS score available for this analysis. This included 12030 RSAs, 6405 TSAs and 3483 HA. Fig 1 shows the spread of mean OSS scores by implant.

HA patients had the largest proportion with an 'unsatisfactory' OSS score (<29) and had significantly more than both TSA & RSA and RSA had significantly more than TSA (p<0.001) (Table 1). In those with an unsatisfactory function, RSA patients were significantly older than both TSA & HA (p<0.001) and HA patients were significantly older than TSA patients (p=0.023). There was no significant difference in gender. RSA patients had a significantly worse ASA than both TSA & HA (p<0.001) indicating poorer pre-operative health status. HA had a significantly worse ASA than TSA (p=0.004). On pairwise comparison of revision, RSA had a significantly lower percentage revised when patients had an unsatisfactory outcome than in both TSA & HA (p<0.001). There was no significant difference in revision in TSA & HA (p=0.018). The lowest 25% of patients had an OSS score of <28 in RSA, <34 in TSA and <23 in HA. The demographics between groups are displayed in Table 2. RSA patients were significantly older than both TSA & HA patients. There was no

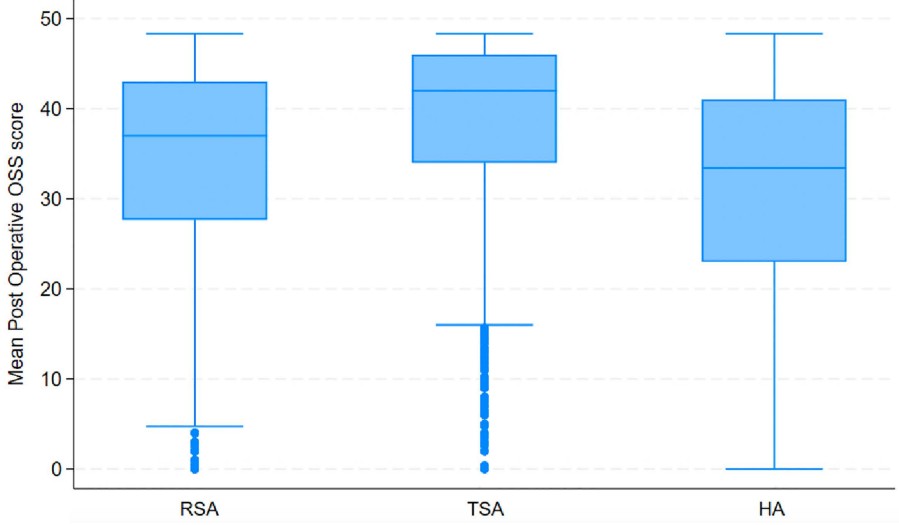

**Fig 1. Mean post operative Oxford shoulder score by implant type.**



**Table 1. Cohort with mean post op score <29.**

| | RSA (n = 3247) | TSA (n = 983) | HA (n = 1328) | Statistical comparison (p-value) |
|---|---|---|---|---|
| Patients with an OSS < 29 (%) | 26.99 | 15.35 | 38.13 | P < 0.001 |
| Age at Shoulder Arthroplasty (SD) | 75.33 (7.90) | 68.97 (10.07) | 69.21 (12.07) | P < 0.001 |
| Gender (%) | | | | |
| Male | 791 (24.36) | 256 (26.04) | 363 (27.33) | 0.094 |
| Female | 2456 (75.64) | 727 (73.96) | 964 (72.59) | |
| Missing | | | 1 (0.08) | |
| ASA | | | | |
| I | 102 (3.14) | 63 (6.41) | 104 (7.83) | P < 0.001 |
| II | 1767 (54.42) | 635 (64.60) | 767 (57.76) | |
| III | 1322 (40.71) | 281 (28.59) | 443 (33.36) | |
| IV | 56 (1.72) | 4 (0.41) | 14 (1.05) | |
| Proportion Revised (%) | 4.87% | 10.58% | 13.86% | p < 0.001 |

SD: Standard Deviation.

**Table 2. Demographics & outcome of those in lowest 25% of function.**

| | RSA (n = 3010) | TSA (n = 1506) | HA (n = 852) | Statistical comparison (p-value) |
|---|---|---|---|---|
| Age at Shoulder Arthroplasty (SD) | 75.28 (7.91) | 69.25 (9.89) | 68.83 (12.19) | P < 0.001 |
| Gender (%) | | | | |
| Male | 739 (24.55) | 393 (26.10) | 238 (27.93) | P < 0.045 |
| Female | 2271 (75.45) | 1113 (73.90) | 613 (71.95) | |
| Missing | | | 1 (0.12) | |
| ASA | | | | |
| I | 95 (3.16) | 101 (6.71) | 68 (7.98) | p < 0.001 |
| II | 1626 (54.02) | 991 (65.80) | 462 (54.23) | |
| III | 1236 (41.06) | 407 (27.09) | 312 (36.62) | |
| IV | 53 (1.76) | 6 (0.40) | 10 (1.17) | |
| Proportion Revised (%) | 4.78% | 8.76% | 15.02% | P < 0.001 |

significant difference in age between TSA & HA patients. There was a significant difference in gender overall however on pairwise testing none of the implants showed significant difference in gender. RSA patients had a significantly worse ASA than both TSA & HA (p < 0.001). HA had a significantly worse ASA than TSA (p < 0.001). RSA had a significantly lower proportion of patients revised than both TSA & HA (p < 0.001). TSA patients within the lowest 25% of function had significantly less revisions than HA however it is important to note the OSS score cut offs for both implants at this level.

The lowest 10% of patients had an OSS score of <18 in RSA, < 24 in TSA and <14 in HA. The demographics between groups are displayed in Table 3. On pairwise analysis of age RSA patients were significantly older than TSA & HA (p < 0.001) and there was no difference between HA and TSA (p = 1.00). There was no significant difference in gender between groups. In terms of ASA, RSA and HA had significantly worse ASA then TSA (p < 0.001 & p = 0.001 respectively). There was no difference between TSA and HA (p = 0.117). RSA had the lowest proportion of those revised within the poorest functioning patients and this was significantly lower than both TSA & HA (p < 0.001). There was no significant difference in proportion revised in TSA & HA (p = 0.056).

**Table 3. Demographics & outcome of those in lowest 10% of function.**

|  | RSA (n = 1195) | TSA (n = 611) | HA (n = 317) | Statistical comparison (p-value) |
|---|---|---|---|---|
| Age at Shoulder Arthroplasty (SD) | 74.89 (8.38) | 68.66 (10.46) | 68.32 (12.47) | P < 0.001 |
| Gender (%) |  |  |  |  |
| Male | 288 (24.10) | 156 (25.53) | 90 (28.39) | P = 0.284 |
| Female | 907 (75.90) | 455 (74.47) | 227 (71.61) |  |
| ASA |  |  |  |  |
| I | 36 (3.01) | 39 (6.38) | 18 (5.68) | P < 0.001 |
| II | 613 (51.30) | 392 (64.16) | 164 (51.74) |  |
| III | 519 (43.43) | 176 (28.81) | 130 (41.01) |  |
| IV | 27 (2.26) | 4 (0.65) | 5 (1.58) |  |
| Proportion Revised (%) | 6.53% | 12.44% | 17.03% | p < 0.001 |

On a sensitivity analysis for patients that underwent their arthroplasty for an elective indication, the results were similar to the primary analysis (Appendix 1). In all measures of poor function (<OSS of 29, lowest 25th percentile and lowest 10th percentile), RSA has a significantly lower percentage revised in comparison to both TSA & HA (<0.001). In addition to the primary analysis in the elective cohort, in unsatisfactory function (OSS < 29) HA had significantly more patients revised compared to TSA (p = 0.003).

Of the 67920 verified implants in the NJR database 2596 resulted in a revision. Of those revised implants 611 had at least one post operative OSS score recorded prior to a revision procedure (Fig 2). In the NJR database 1081 RSAs were revised (2.79%), 686 TSAs (3.75%) and 819 (7.52%) HAs.

Demographic comparison between groups are shown in Table 4. Revision patients that required a revision of a TSA and RSA were significantly older than those receiving a HA (p = 0.042 and p=<0.001 respectively). Patients receiving a revision of an RSA were also significantly older than those receiving a revision of a TSA (p = 0.017). In terms of gender, patients requiring a revision of a TSA and HA were more likely to be female than if you required a revision of an RSA (p = 0.008 and p = 0.002 respectively). There was no difference in the gender distribution between TSA and HA (p = 0.785). The median time to revision in the cohort was 3.1 years (IQR 1.6–5.1). On pairwise testing RSA was revised significantly sooner than TSA and HA (p=<0.001 and p = 0.006 respectively) but there was no significant difference in revision time between TSA and HA (p = 0.152).

The median most recent OSS score prior to revision for RSA was 29 (IQR 16–39), for TSA was 30 (IQR 21–41) and for HA was 23 (15–32) (Fig 3). Kruskall Wallis H test showed a significant difference between the groups (p < 0.001). On pairwise analysis there was a significant difference between the OSS pre revision in HA compared to both TSA (p < 0.001) and RSA (p = 0.002). There was however no difference between TSA and RSA (p = 0.252).

## Discussion

This study aimed to assess whether low revision rates in RSA were reflective of implant success or are influenced by factors such as surgical decision making and patient suitability for revision. This study utilised PROMs to assess patient function in relation to revision which has previously been used to assess revision thresholds in knee arthroplasty [9,10]. In the NJR in both the UK and in Australia as well as various mid to long term studies, RSA has lower revision rates than TSA and TSA lower than HA [2,11,12]. Comparatively in this study poorly functioning and unsatisfied RSA patients are less likely to be revised than TSA & HA patients. Indication for primary operation did not affect this finding. The cohort of patients receiving an RSA are older and had a higher ASA than TSA patients. When primary RSA went on to revision, they were revised sooner than TSA & HA. The findings of this study are of concern as given the dramatic rise in RSA use and the developing breadth of indications and patient demographic for which it is now being used, if the function of RSA

*Percentage of total revision cases for that implant

OSS: Oxford Shoulder Score

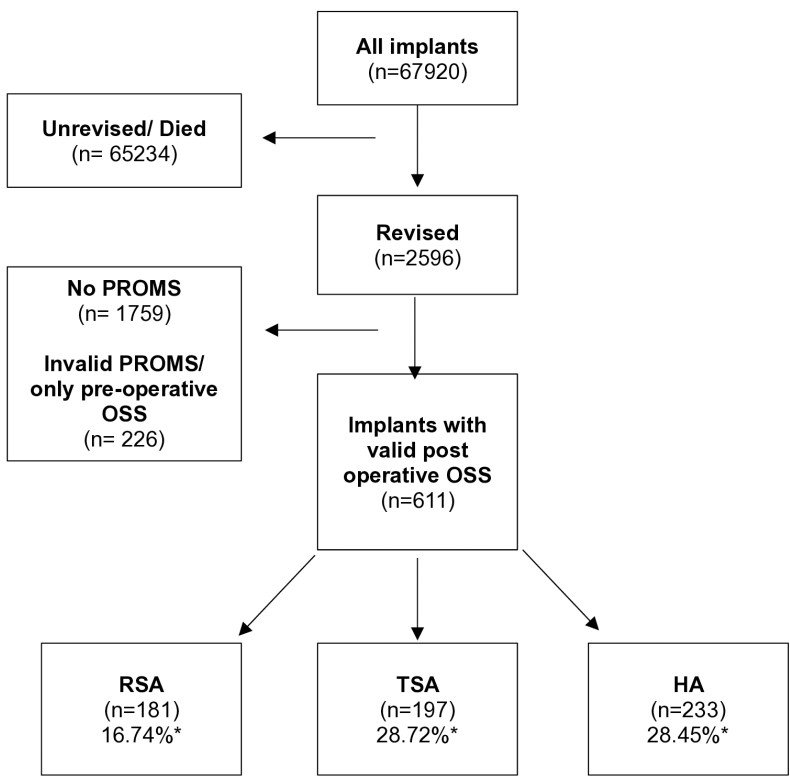

**Fig 2. Flow chart of revised patients cohort selection.**

patients is poor and revision rates are not reflective of this, this may result in a large burden of patients with poor function in the future [1,3]. This poorly functioning cohort are of a broadened demographic to the previously elderly RSA population and therefore may be less accepting of this function. This in turn may further increase the demand for revision surgery.

Overall, HA patients had the highest proportion of unsatisfactory OSS scores and in this study had a significantly worse function prior to revision than TSA and RSA. This finding is also reflected in the NJR report and a comparative study of TSA vs HA in osteoarthritis has also echoed this finding [2,13]. HA is mainly used in younger patients with an intact cuff without glenoid disease and in proximal humeral fractures [2,13]. In this younger demographic (<60) for primary shoulder arthroplasty there is a fourfold increase risk in lifetime revision compared to those greater than 85 and therefore longevity of the primary replacement in this demographic is crucial [14].. This study showed HA patients are revised later than RSA patients. This may be because revision is prolonged, and therefore function may be worse, until it becomes a necessity as if the patient is younger they may require a second revision in their lifetime which is known to have poor outcomes in comparison to primary and revision surgery [15,16]. However lower OSS in those that went onto have a revision in HA patients compared to TSA & HA patients, may be reflective just of HA performance in general given post operatively patients appear less satisfied compared to both RSA & TSA.

**Table 4. Demographics of patients that received a revision by implant type.**

| Demographic | RSA (n = 181) | TSA (n = 197) | HA (n = 233) | Statistical difference |
|---|---|---|---|---|
| Age at Primary Shoulder Arthroplasty (SD) | 71.93 (7.82) | 68.10 (9.27) | 66.29 (10.25) | P < 0.001 |
| Age at Revision Shoulder Arthroplasty (SD) | 74.89 (7.82) | 72.19 (9.38) | 69.95 (10.30) | P < 0.001 |
| Gender (%) | | | | |
| Male | 84 (46.41) | 65 (32.99) | 74 (31.76) | p = 0.004 |
| Female | 97 (53.59) | 132 (67.01) | 159 (68.24) | |
| ASA grade (%) | | | | |
| I | 11 (6.08) | 19 (9.64) | 28 (12.02) | P = 0.236 |
| II | 123 (67.96) | 140 (71.07) | 159 (68.24) | |
| III | 45 (24.86) | 38 (19.29) | 44 (18.88) | |
| IV | 2 (1.10) | 0 (0) | 2 (0.86) | |
| Time to revision (years) (IQR) | 2.4 (1.1-4.2) | 3.5 (2.0-6.0) | 3.4 (1.8-5.2) | p < 0.001 |
| Indication for revision (%)* | | | | |
| 1st | Infection (37.02) | Cuff Deficiency (46.19) | Cuff deficiency (38.20) | |
| 2nd | Instability (15.47) | Aseptic loosening (21.83) | Native Glenoid Wear (28.33) | |
| 3rd | Aseptic Loosening (19.34) | Glenoid Wear (16.75) | Unexplained pain (8.58) | |

*not mutually exclusive IQR: Interquartile Range.

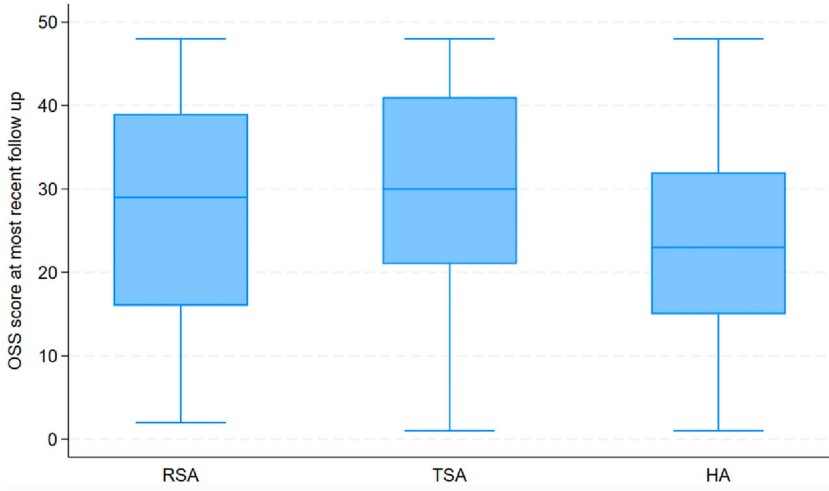

**Fig 3. Box & whisker plot of Oxford shoulder score at most recent follow up prior to revision by implant type.**

RSA in the past 10 years has shown exponential growth and the low reported revision rates may encourage its further increasing use [2]. This study has shown RSA patients are likely to be older when having revision surgery compared to HA and TSA and given the primary demographic being older this is to be expected. There was no difference in gender split between implants at primary surgery, however a higher proportion of males went on to have a revision in RSA compared to HA and TSA (46.41% in RSA, 32.99% in TSA & 31.76% in HA p = 0,004). This gender split was also not reflective of primary RSA procedures where only 24.36% of cases in this study were done in male patients. A recent study aimed to explore outcomes associated with gender in primary RSA and found female gender was a weak negative predictor of



constant score, a marker of shoulder function, and it was theorised that the main reason for the worse outcome in female patients seems to be a combination of higher preoperative disability and higher incidence of fractures [17]. These worsened outcomes post primary surgery may be leading to adversity to perform revision surgery in females due to potentially being higher risk patients with a lower likelihood of clinical success in a revision procedure. Given that the proportion of RSA revised in this study is smaller than the other implants, with a larger proportion of male patients that is not reflective of the primary cohort, this gender difference may show a gender selection bias and adversity to performing a revision procedure in a female patient.

In terms of ASA, in poorly functioning and unsatisfied patients, primary RSA patients had a higher ASA than both TSA & HA indicating poorer a pre-operative health status. However, when looking at the cohort that went onto to revision there was no difference between implants. Again, this possibly suggests a selection bias where higher comorbid status affects patient selection for revision in RSA patients and therefore low revision rates may not be reflective of a cohort of older frailer patients with poor function post operatively.

This hypothesis that revision rates are not reflective of implant performance has also been highlighted in knee arthroplasty [18]. A recent New Zealand joint registry study found that although unicompartmental knees (UKA) had better PROMs, they had higher revision rates compared to total knee replacements (TKA) [9]. The explanation for this was that surgeons may have a lower revision threshold to revise a UKA to a TKA. They also suggested this may be due to additional UKA failure modes compared to TKA such as osteoarthritis progression due to uncovered joint surfaces and bearing dislocations [9]. This is similar to this study where at each functional threshold TSA has better functional scores than RSA however TSA has a higher proportion of revised implants. The most common cause of revision in TSA was cuff deficiency which is an additional mode of failure not present for RSA. Therefore similar to UKA, surgeons have a lower threshold to revise a TSA with a reasonable salvage option of an RSA and the higher rates may be reflective of the additional failure mechanism of cuff failure [19].

The limitations of this study centre around the availability of PROMs data. Despite PROMs data collection being mandatory from the NJR, 32% had a valid post operative PROMs recorded and <30% of cases those that resulted in revision had valid post operative OSS scores. The poor collection of PROMS is consistent across implants and therefore comparatively differences within groups may not be as affected. The NJR also does not collect PROMs in revision cases pre or post operatively nor does it investigate reason for non-revision in poorly functioning patients. This may be an opportunity missed to establish the true success of these implants and predict for a potential increasing revision burden given highlighted in this study. In the revision analysis although the patient groups exceeded over 180 patients', interpretation should be cautious as this may not be reflective of the entire revision population. This study however is the first to attempt to look at if there is a selection bias dependant on implant for revision shoulder arthroplasty.

## Conclusion

Revision rates of RSA are lower than that of HA and TSA however this study suggests that this may not be truly reflective of the implants success as patients with poor function in RSA are less likely to be revised than those with poor function in both TSA & HA. There is a possible channelling bias where female patients and those with higher co-morbidities are not revised. As RSA use expands, its assumed low revision rate must be reassessed to prevent long-term burdens of poorly functioning implants. Further research is needed to determine whether surgical selection bias influences revision rates and to establish additional benchmarks or surrogates in joint registries for a more comprehensive assessment of implant performance.

## Acknowledgments

We thank the NJR research committee, and staff at the NJR, for facilitating this work. The authors have conformed to the NJR's standard protocol for data access and publication. The views expressed represent those of the authors and do not necessarily reflect those of the NJR steering committee, research subcommittee, or HQIP.

## Author contributions

**Conceptualization:** Olivia O'Malley, Sanjeeve Sabharwal, Peter Reilly.

**Data curation:** Olivia O'Malley.

**Formal analysis:** Olivia O'Malley, Andrew Davies.

**Funding acquisition:** Olivia O'Malley, Sanjeeve Sabharwal, Peter Reilly.

**Investigation:** Olivia O'Malley.

**Methodology:** Olivia O'Malley, Andrew Davies, Amar Rangan, Sanjeeve Sabharwal, Peter Reilly.

**Project administration:** Peter Reilly.

**Resources:** Peter Reilly.

**Supervision:** Amar Rangan, Sanjeeve Sabharwal, Peter Reilly.

**Writing – original draft:** Olivia O'Malley.

**Writing – review & editing:** Andrew Davies, Amar Rangan, Sanjeeve Sabharwal, Peter Reilly.

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
