## [Decision Letter · Decision Letter 0]

21 Jul 2025

PONE-D-25-30411Is there a difference in thresholds for revision between shoulder arthroplasty types? A National Joint Registry StudyPLOS ONE?

Dear Dr. O'Malley,

Thank you for submitting your manuscript to PLOS ONE. After careful consideration, we feel that it has merit but does not fully meet PLOS ONE’s publication criteria as it currently stands. Therefore, we invite you to submit a revised version of the manuscript that addresses the points raised during the review process.

We look forward to receiving your revised manuscript.

Kind regards,

Nan Jiang

Academic Editor

PLOS ONE

3. Thank you for stating the following financial disclosure: [The authors disclose receipt of the following financial or material support for the research, authorship, and/or publication of this article: an institutional British Elbow and Shoulder Society pump primer research grant, and a Royal College of Surgeons Research Fellowship.]. 

4.  Thank you for stating the following in the Competing Interests section: [O.O’Malley is an RCS England Research Fellow funded by The Arthritis Research Trust Research Fellowship with support from the Rosetrees Trust. A. Rangan reports an institutional grant from NIHR and AO UK&I, as well as research and educational grants from DePuy J&J Ltd, unrelated to this study. A. Rangan is also an Elected Trustee of the British Orthopaedic Association and is on the funding committee of NIHR i4i.].

6. In the online submission form, you indicated that [Data cannot be shared publicly because of data is held by the NJR. Data are available from the NJR on request.].

Reviewers' comments:

Reviewer's Responses to Questions

**Comments to the Author**

1. Is the manuscript technically sound, and do the data support the conclusions?

Reviewer #1: Partly

Reviewer #2: Yes

2. Has the statistical analysis been performed appropriately and rigorously?

Reviewer #1: Yes

Reviewer #2: Yes

3. Have the authors made all data underlying the findings in their manuscript fully available?

Reviewer #1: No

Reviewer #2: No

4. Is the manuscript presented in an intelligible fashion and written in standard English?

Reviewer #1: No

Reviewer #2: Yes

Reviewer #1: General Comments

• The authors present an analysis that could be a valuable contribution to literature in the area of shoulder arthroplasty. Their takeaway that lower revision rates for RSA compared to alternative procedures may not indicate increased success of the RSA procedure is important to know. However, this manuscript lacks substantial detail to support its conclusions as written.

• Of particular concern are the analyses related to the “lowest 25%” and “lowest 10%” of patients with each implant type. These patients have different OSS scores pre-operatively across implant types making it hard to understand the relevance of all other subsequent comparisons (such as proportion revised). I recommend clearly explaining the purpose of these extra analyses (instead of just presenting the analysis of OSS <29) in the introduction/methods and specifically addressing how the reader should interpret these findings in the discussion. Without clear rationale for their inclusion, this manuscript would be strengthened by removing the “lowest 25%” and “lowest 10%” analyses.

• This manuscript contains many typos and punctuation errors. At several points in the manuscript, these errors greatly impact readability. These errors are so numerus that I have not included edits in this review. Given that PLOS ONE does not copyedit, I recommend the authors seek editorial help prior to submitting a revision.

• The manuscript lacks line numbers as indicted in author instructions. This makes it particularly difficult to provide corrections for typos and grammatical errors.

• Figure captions are missing

Abstract

• Methods – there is a reference to analysis of the lowest 25% and the lowest 10%. Please add clarifying language – the lowest 25% of ___ and the lowest 10% of ___.

• The methods need to explain that this was a database search over X years

• Please define NJR prior to using it.

Introduction

• In general, this introduction lacks structure and sufficient background information. Consider starting more broadly, perhaps explaining why these arthroplasties are needed and why high rates of revision are of concern. Use citations to support statements of changing rates of procedure types. Explain the gap in knowledge that your study aims to fill then follow with your purpose statement and hypothesis.

• Citations 1-3 are about outcomes and do not reference NJR. Please replace with appropriate citations.

• “attributed to its increased scope of use” – please include citations and briefly describe this increased scope

• “low reported revision rates” – please include citations

• PROMs – please define before use; There is insufficient information preceding this sentence for the reader to understand why PROMs are needed. See first comment under Introduction.

• Please explain what is meant by “threshold”

Methods

• Define PROMs in the intro then use the acronym here

• “12 scored questions which are scored a system scoring from 0 to 4” – this statement is missing a word and can be cleaned up for clarity. Perhaps “12 questions which are each scored from 0 to 4”

• It would be helpful to 1) state the types of questions in OSS (pain ratings, function/disability ratings, both?) and 2) indicate what a maximum score of 48 means (e.g., no pain or no disability)

• “As defined by Dawson et. al…” needs a citation.

• “This method was implored in this study” – please expand to be clear about to what you are referring. Does “this method” refer to the scoring procedure? Are OSS total scores available directly from NJR or were you calculating for each patient?

• I would recommend using patient first language in this paragraph. For example: “The primary outcome was to compare, at differing OSS score cutoffs, the percentage of patients with each implant type who underwent revision surgery.”

• As indicted above, “threshold for revision” should be more clearly defined

• “selected” should be “selecting”

• Please clarify your secondary outcome comparison; are you comparing patients across implant types or comparing patients with revision to those without revision?

• Statistical analysis section – see comment above about patient first language

• “The mean OSS at follow-up for each patient was calculated” belongs earlier in the methods section. At which follow-up timepoint was this done?

• “The outcome of these patients in terms of those that went on to revision was calculated and described with descriptive statistics” – this sentence is confusing as written. Is this saying that the rates of revision for each implant type were calculated and compared descriptively? Please revise for clarity.

• Last sentence of this paragraph – this is the first we have heard about elective indication. Please provide background for this in your intro/methods and explain what the alternative to elective indication would be.

Results

• Please explain what “validated implants” are in the methods

• Figures 1-3 – these data would be more helpfully displayed in a single plot so the reader can easily compare mean OSS scores across implant types.

• How far post-operatively were these OSS scores taken? The methods mention OSS at 6 months, 3 years, and 5 years.

• If you evaluated the rates of patients being biologically male vs. female, the more appropriate term to use throughout this manuscript is “sex” instead of “gender”.

• Define ASA. This should be included in the methods if you are going to include in results.

• What should we take away from the fact that the lowest 25% OSS score was different across the implant types? Were these significantly different from one another? If so, what is the point of this analysis? See the comment under General Comments.

• “There was a minimally significant difference” – the significant difference was identified and shouldn’t be quantified as minimal here.

• Sensitivity analysis paragraph – “HA had significantly more patients revised to TSA” – should this say “more patients revised compared to TSA”?

Discussion

• Sentence: “This is of concern as given the dramatic rise in RSA use and the developing breadth of indications and patient demographic for which it is now being used, if the function of the current patients where the RSA has been used is poor and revision rates are not reflective of this, this may result in a large burden of patients with poor function in the future that are less accepting than the original demographic, increasing the demand for revision surgery.”

o This should be revised for typos and split into multiple sentences. Statements require citations.

o What is meant by “are less accepting”? How is this supported by this study’s findings?

• Sentence: “Therefore, poorer pre revision function may be due to prolonging revision, which in this study HA patients are shown to have a revision later than RSA patients, until it becomes a necessity as if the patient is younger they may require a second revision in their lifetime which is known to have poor outcomes in comparison to primary and revision surgery (13,14).”

o This sentence is long and difficult to follow. Please split up for improved clarity.

• Sentence: However, this lower OSS in revision cases may be reflective just of HA performance in general given post operatively patients appear less satisfied rather than a higher threshold for revision in HA compared to TSA and RSA.”

o This sentence is also confusing as written. I believe you are trying to say that post-revision OSS scores may be lower in HA than other implants because pre-revision OSS scores are lower in HA, however it is not clear if this what is meant. Please use clarifying language.

• ASA paragraph – explain what a higher ASA means

• Sentence: “The explanation for this was that surgeons may have a lower revision threshold to revise a UKA to a TKA and the possible association additional UKA failure modes compared to TKA such as osteoarthritis progression due to uncovered joint surfaces and bearing dislocations which are more common for UKA (6).”

o Long and confusing as written, please split up/revise for clarity

Tables

• Table 1

o What does % of PROMS cohort mean?

o Age at primary what?

o Please define all terms and include a caption

o Please explain what statistical comparison’s p-value is provided. If it is for the overall Kruskall-Wallace or Chi Squared test, please use indicators (such as superscript letters) to indicate which pairwise comparisons were significant as well.

• Table 2 and 3

o See comments from Table 1

• Table 4

o See comments from Table 1

o Define IQR

o Provide a more descriptive title

Figures

• All figures need captions including definitions of all abbreviations

• Figure 5 – add indicators of statistically significant implant differences

Reviewer #2: Dear Authors

Thank you for your submission. This is a large data set and analysed well. The results and interpretation fit with the knee literature, which in effect, shows that a uni knee is easier to revise and, therefore, gets revised more often than a TKR.

There are some small typos that need correcting. I think the tables and the PRISMA flow sheet need some work to make them more presentable. Please work on this.

Also, please provide a statement on consent of the patients signing up to the registry and their PROMS data being used.

Otherwise, the manuscript is sound and I will recommend it for publication with minor changes

**Do you want your identity to be public for this peer review?** For information about this choice, including consent withdrawal, please see our Privacy Policy

Reviewer #1: No

Reviewer #2: **Yes: ** Oliver Boughton

---

## [Author Response · Author response to Decision Letter 1]

23 Jul 2025

Dear Editor,

Thank you for the opportunity to revise out manuscript. I attach our revised manuscript with responses to reviewers’ comments:

Reviewer #1

Comment:

The authors present an analysis that could be a valuable contribution to literature in the area of shoulder arthroplasty. Their takeaway that lower revision rates for RSA compared to alternative procedures may not indicate increased success of the RSA procedure is important to know. However, this manuscript lacks substantial detail to support its conclusions as written.

Response:

We thank you for the reviewers’ comment and are grateful for the acknowledgement that this could be of valuable contribution to the literature. We hope with the revisions presented sufficient detail is now present for publication

Comment:

Of particular concern are the analyses related to the “lowest 25%” and “lowest 10%” of patients with each implant type. These patients have different OSS scores pre-operatively across implant types making it hard to understand the relevance of all other subsequent comparisons (such as proportion revised). I recommend clearly explaining the purpose of these extra analyses (instead of just presenting the analysis of OSS <29) in the introduction/methods and specifically addressing how the reader should interpret these findings in the discussion. Without clear rationale for their inclusion, this manuscript would be strengthened by removing the “lowest 25%” and “lowest 10%” analyses.

Response:

These secondary cut-off scores were intentionally selected to reflect clinically meaningful thresholds of poor function within the context of the known post-operative performance characteristics of different shoulder implants. As practicing shoulder surgeons are aware, the expected functional outcomes vary significantly between implant types (e.g., reverse vs anatomical TSA), and a universal threshold could risk misclassification. Our method provides an additional clinically relevant comparison of poorly functioning implants by adjusting for these expected differences. We have clarified this rationale in the Methods section, and the implications for interpreting implant performance are now explicitly discussed in the Discussion. We believe this approach will be of direct interest to shoulder surgeons, as it aligns with how outcomes are interpreted in real-world clinical decision-making

Comment:

This manuscript contains many typos and punctuation errors. At several points in the manuscript, these errors greatly impact readability. These errors are so numerus that I have not included edits in this review. Given that PLOS ONE does not copyedit, I recommend the authors seek editorial help prior to submitting a revision.

Response:

The manuscript has been further proofread by the corresponding and senior authors to ensure these errors have been corrected.

Comment:

The manuscript lacks line numbers as indicted in author instructions. This makes it particularly difficult to provide corrections for typos and grammatical errors.

Response:

Line numbers were included and we are unsure why this cannot be seen in the original submission. We will ensure they are present in the revision.

Comment:

Figure captions are missing

Response:

Again, we are unsure why figure captions are not present in the submission as they were included however, we will ensure they are present for the revision.

Abstract

Comment

• Methods – there is a reference to analysis of the lowest 25% and the lowest 10%. Please add clarifying language – the lowest 25% of ___ and the lowest 10% of ___.

• The methods need to explain that this was a database search over X years

• Please define NJR prior to using it.

Response:

These corrections have all now been added to the abstract.

Introduction

Comment:

In general, this introduction lack’s structure and sufficient background information. Consider starting more broadly, perhaps explaining why these arthroplasties are needed and why high rates of revision are of concern. Use citations to support statements of changing rates of procedure types. Explain the gap in knowledge that your study aims to fill then follow with your purpose statement and hypothesis.

Response:

The introduction has now been restructured, expanded and citations added.

Comment:

Citations 1-3 are about outcomes and do not reference NJR. Please replace with appropriate citations.

Response:

Corrected

Comment:

“attributed to its increased scope of use” – please include citations and briefly describe this increased scope

Response:

A reference and sentence for how RSA is now being used and a reference has been added.

Comment:

“low reported revision rates” – please include citations

Response:

Added

Comment:

PROMs – please define before use; There is insufficient information preceding this sentence for the reader to understand why PROMs are needed. See first comment under Introduction. Please explain what is meant by “threshold”

Response:

This has now been clarified and added to the introduction

Methods

Comment:

Define PROMs in the intro then use the acronym here

Response

Done

Comment:

“12 scored questions which are scored a system scoring from 0 to 4” – this statement is missing a word and can be cleaned up for clarity. Perhaps “12 questions which are each scored from 0 to 4”

Response:

This has now been reworded

Comment:

It would be helpful to 1) state the types of questions in OSS (pain ratings, function/disability ratings, both?) and 2) indicate what a maximum score of 48 means (e.g., no pain or no disability)

Response:

This sentence has now been added to the methods explaining the score further.

Comment:

“As defined by Dawson et. al…” needs a citation.

Response:

Added

Comment:

“This method was implored in this study” – please expand to be clear about to what you are referring. Does “this method” refer to the scoring procedure? Are OSS total scores available directly from NJR or were you calculating for each patient?

Response:

This has now been clarified in the methods.

Comment:

I would recommend using patient first language in this paragraph. For example: “The primary outcome was to compare, at differing OSS score cutoffs, the percentage of patients with each implant type who underwent revision surgery.”

As indicted above, “threshold for revision” should be more clearly defined

“selected” should be “selecting”

Please clarify your secondary outcome comparison; are you comparing patients across implant types or comparing patients with revision to those without revision?

Response:

The language has been altered with clearer definitions and outcomes specified.

Comment

Statistical analysis section – see comment above about patient first language

“The mean OSS at follow-up for each patient was calculated” belongs earlier in the methods section. At which follow-up timepoint was this done?

“The outcome of these patients in terms of those that went on to revision was calculated and described with descriptive statistics” – this sentence is confusing as written. Is this saying that the rates of revision for each implant type were calculated and compared descriptively? Please revise for clarity.

Last sentence of this paragraph – this is the first we have heard about elective indication. Please provide background for this in your intro/methods and explain what the alternative to elective indication would be.

Response:

The language has been altered in this section and time points specified. The sentence regarding percentage revisions has been clarified and simplified for ease of reading. The introduction as previously been discussed has been expanded discussing indications and this final sentence has now been clarified.

Results:

Comment:

Please explain what “validated implants” are in the methods

Response:

This has now been explained in the methods.

Comment:

Figures 1-3 – these data would be more helpfully displayed in a single plot so the reader can easily compare mean OSS scores across implant types.

Response:

This has now been converted into a box and whisker plot for ease of viewing and interpretation

Comment

How far post-operatively were these OSS scores taken? The methods mention OSS at 6 months, 3 years, and 5 years.

Response

As specified in the method this is a mean score across all the follow up periods

Comment:

If you evaluated the rates of patients being biologically male vs. female, the more appropriate term to use throughout this manuscript is “sex” instead of “gender”.

Response:

The reporting of gender here is consistent with the study protocol as approved by the NJR research committee and falls in line with other observational research reported from the NJR. An example of this: Penfold CM, Whitehouse MR, Blom AW, Judge A, Wilkinson JM, Sayers A. A comparison of comorbidity measures for predicting mortality after elective hip and knee replacement: A cohort study of data from the National Joint Registry in England and Wales. PLoS One. 2021 Aug 12;16(8):e0255602. doi: 10.1371/journal.pone.0255602. PMID: 34383814; PMCID: PMC8360555.

Comment:

Define ASA. This should be included in the methods if you are going to include in results.

Response:

Done

Comment:

What should we take away from the fact that the lowest 25% OSS score was different across the implant types? Were these significantly different from one another? If so, what is the point of this analysis? See the comment under General Comments.

Response:

This has been discussed in the discussion paragraph 2.

Comment

“There was a minimally significant difference” – the significant difference was identified and shouldn’t be quantified as minimal here.

Response

This has been edited

Comment:

Sensitivity analysis paragraph – “HA had significantly more patients revised to TSA” – should this say, “more patients revised compared to TSA”?

Response

This has been corrected

Discussion

Comment:

Sentence: “This is of concern as given the dramatic rise in RSA use and the developing breadth of indications and patient demographic for which it is now being used, if the function of the current patients where the RSA has been used is poor and revision rates are not reflective of this, this may result in a large burden of patients with poor function in the future that are less accepting than the original demographic, increasing the demand for revision surgery.”

This should be revised for typos and split into multiple sentences. Statements require citations.

Response:

This has now been rewritten and citations added.

Comment:

What is meant by “are less accepting”? How is this supported by this study’s findings?

Response:

This has now been explained further – younger patients now receiving an RSA will likely require higher function of their prosthesis compared to the elderly patients who were the patients traditionally receiving an RSA.

Comment:

Sentence: “Therefore, poorer pre revision function may be due to prolonging revision, which in this study HA patients are shown to have a revision later than RSA patients, until it becomes a necessity as if the patient is younger they may require a second revision in their lifetime which is known to have poor outcomes in comparison to primary and revision surgery (13,14).”

This sentence is long and difficult to follow. Please split up for improved clarity.

Response:

This has been reworded.

Comment:

Sentence: However, this lower OSS in revision cases may be reflective just of HA performance in general given post operatively patients appear less satisfied rather than a higher threshold for revision in HA compared to TSA and RSA.”

This sentence is also confusing as written. I believe you are trying to say that post-revision OSS scores may be lower in HA than other implants because pre-revision OSS scores are lower in HA, however it is not clear if this what is meant. Please use clarifying language.

Response:

This has been reworded.

Comment:

• ASA paragraph – explain what a higher ASA means

Response

Completed

Comment:

Sentence: “The explanation for this was that surgeons may have a lower revision threshold to revise a UKA to a TKA and the possible association additional UKA failure modes compared to TKA such as osteoarthritis progression due to uncovered joint surfaces and bearing dislocations which are more common for UKA (6).” Long and confusing as written, please split up/revise for clarity

Response:

This has been separated and revised.

Tables

Comment:

Table 1

o What does % of PROMS cohort mean?

o Age at primary what?

o Please define all terms and include a caption

o Please explain what statistical comparison’s p-value is provided. If it is for the overall Kruskall-Wallace or Chi Squared test, please use indicators (such as superscript letters) to indicate which pairwise comparisons were significant as well.

Response

PROMS cohort has been changed to be clearer as has age at primary. All terms have now been explained in prior text or in table caption. The pairwise analyses have all been specified in the text and statistical tests were specified in the methods.

• Table 2 and 3

o See comments from Table 1

• Table 4

o See comments from Table 1

o Define IQR

o Provide a more descriptive title

Response:

All tables have now been edited as for table 1 and captions/ titles expanded.

Figures

Comment:

• All figures need captions including definitions of all abbreviations

• Figure 5 – add indicators of statistically significant implant differences

Response:

Figure captions have been changed and abbreviations added

Reviewer #2:

Dear Authors,

Thank you for your submission. This is a large data set and analysed well. The results and interpretation fit with the knee literature, which in effect, shows that a uni knee is easier to revise and, therefore, gets revised more often than a TKR.

Comment:

There are some small typos that need correcting. I think the tables and the PRISMA flow sheet need some work to make them more presentable. Please work on this.

Response:

The tables and flow chart have been edited to be more understandable and clearer.

Comment:

Also, please provide a statement on consent of the patients signing up to the registry and their PROMS data being used.

Response:

This has been added to the methods.

Comment:

Otherwise, the manuscript is sound and I will recommend it for publication with minor changes

Response:

We thank you for your review and comments.

We look forward to your response

Yours Sincerely

Miss O. O’Malley

PhD Candidate & Trauma & Orthopaedic Registrar Imperial College London

---

## [Decision Letter · Decision Letter 1]

10 Aug 2025

Is there a difference in thresholds for revision between shoulder arthroplasty types? A National Joint Registry Study

PONE-D-25-30411R1

Dear Dr. O'Malley,

We’re pleased to inform you that your manuscript has been judged scientifically suitable for publication and will be formally accepted for publication once it meets all outstanding technical requirements.

Kind regards,

Nan Jiang

Academic Editor

PLOS ONE

Reviewers' comments:

Reviewer's Responses to Questions

**Comments to the Author**

Reviewer #2: All comments have been addressed

2. Is the manuscript technically sound, and do the data support the conclusions?

Reviewer #2: Yes

3. Has the statistical analysis been performed appropriately and rigorously?

Reviewer #2: Yes

4. Have the authors made all data underlying the findings in their manuscript fully available?

Reviewer #2: Yes

5. Is the manuscript presented in an intelligible fashion and written in standard English?

Reviewer #2: Yes

Reviewer #2: All comments addressed, thank you, I recommend this for publication. This article significantly contributes to the literature

**Do you want your identity to be public for this peer review?** For information about this choice, including consent withdrawal, please see our Privacy Policy

Reviewer #2: **Yes: ** Oliver Boughton

---

## [Editor Report · Acceptance letter]

PONE-D-25-30411R1

PLOS ONE

Dear Dr. O'Malley,

I'm pleased to inform you that your manuscript has been deemed suitable for publication in PLOS ONE. Congratulations! Your manuscript is now being handed over to our production team.

Kind regards,

on behalf of

Dr. Nan Jiang

Academic Editor

PLOS ONE